# Association Between Dietary Tomato Intake and Blood Eosinophil Count in Middle-Aged and Older Japanese Individuals: A Population-Based Cross-Sectional Study [note 1]

**DOI:** 10.3390/nu17213467

**Published:** 2025-11-03

**Authors:** Akinori Hara, Hiromasa Tsujiguchi, Rio Fukuchi, Masaharu Nakamura, Jam Camara, Marama Talica, Jiaye Zhao, Chie Takazawa, Fumihiko Suzuki, Haruhiko Ogawa, Takayuki Kannon, Takehiro Sato, Atsushi Tajima, Hiroyuki Nakamura

**Affiliations:** 1Department of Hygiene and Public Health, Faculty of Medicine, Institute of Medical, Pharmaceutical and Health Sciences, Kanazawa University, Kanazawa 920-8640, Japan; t-hiromasa@med.kanazawa-u.ac.jp (H.T.); cjam949@gmail.com (J.C.); h-ogawa@jintikai.com (H.O.); hnakamu@staff.kanazawa-u.ac.jp (H.N.); 2Department of Bioinformatics and Genomics, Faculty of Medicine, Institute of Medical, Pharmaceutical and Health Sciences, Kanazawa University, Kanazawa 920-8640, Japantakayuki.kannon@fujita-hu.ac.jp (T.K.); tsato@cs.u-ryukyu.ac.jp (T.S.); atajima@med.kanazawa-u.ac.jp (A.T.); 3Department of Geriatric Dentistry, Ohu University School of Dentistry, Koriyama 963-8041, Japan; f-suzuki@den.ohu-u.ac.jp; 4Department of Internal Medicine, Kanazawa Kasuga Clinic, Kanazawa 920-0036, Japan

**Keywords:** food, nutrition, allergy, prevention, polygenic risk score

## Abstract

**Background/Objectives**: Although tomato consumption has been associated with positive health outcomes, it remains unclear whether it can prevent or exacerbate allergic diseases by regulating eosinophils. We explored the association between dietary tomato intake and blood eosinophil counts in Japanese individuals. **Methods**: This population-based, cross-sectional study included 1013 participants aged ≥ 40 years. The dietary intake of tomatoes was assessed using a validated, self-administered diet history questionnaire. The peripheral blood eosinophil count was measured, and an elevated blood eosinophil count was defined as a value that exceeded the ≥75th percentile. **Results**: The mean age of the participants was 62.5 ± 11.2 years, with 474 (46.8%) being male. Overall, 252 participants exhibited elevated blood eosinophil counts (≥204/μL). In the multivariable logistic regression model with adjustment for potential confounders, an increase in tomato intake of 10 g was inversely associated with an elevated blood eosinophil count (odds ratio [OR], 0.895; 95% confidence interval [CI], 0.834–0.961). Except for chronic kidney disease, the baseline participant characteristics did not influence this association. **Conclusions**: Low dietary tomato intake was associated with an elevated blood eosinophil count in middle-aged and older Japanese individuals. These results may provide insight into the dietary management of eosinophil-related allergic and type 2 inflammatory diseases.

## 1. Introduction

Allergy is a hypersensitivity reaction that is induced by immunological mechanisms. In addition to the risk of potentially fatal anaphylaxis, a wide variety of allergic symptoms can adversely impact an individual’s quality of life [1]. In 2021, the National Center for Health Statistics of the United States reported that 31.8% of adults aged 18 and older experienced seasonal allergies, eczema, or food allergies [1]. Additionally, it is estimated that approximately two-thirds of the Japanese population are affected by allergic diseases, with allergic rhinitis being the most prevalent [2].

Blood or tissue eosinophils, which are a type of bone marrow-derived granulocytes, are the pathogenic drivers of type 2 hypersensitivity reactions that underlie allergies. Their levels are often quantified and used as dominant and practical clinical biomarkers. Eosinophil concentrations in peripheral blood are particularly useful in diagnosing and treating eosinophil-associated diseases such as hypereosinophilic syndrome, asthma, and eosinophilic granulomatosis with polyangiitis (EGPA) [3]. Blood eosinophils are used as a therapeutic target and disease activity indicator in such disorders [3,4]. Furthermore, in addition to EGPA, peripheral blood eosinophil counts have recently been incorporated into the diagnostic criteria for drug-induced hypersensitivity syndrome/drug reaction with eosinophilia and systemic symptoms [5].

In addition to the principal lifestyle modification of avoiding triggers, it is critical to implement measures to control allergic reactions for preventing and alleviating symptoms of allergic diseases. Besides their nutritive value, various foods have been reported to possess immunomodulatory properties [6]. From a mechanistic insight, it is increasingly recognized that in general, the interaction between the environment and genetic background, often mediated by the epigenetic mechanisms, is involved in exerting the dietary effects on allergies development [7]. Tomatoes are a widely consumed vegetable that contain a variety of nutrients, such as minerals, vitamins, polyphenols, and carotenoids, and are known to have inhibitory effects on immune and inflammatory responses [8]. Increasing evidence has demonstrated that tomato consumption is associated with reduced risk of mortality, cardiovascular disease (CVD), and certain cancers, and these beneficial effects may be mediated by the antioxidant, anti-inflammatory, and anti-cancer properties of the tomato constituents, including lycopene, and polyphenols [8,9]. Flavonoids, a class of polyphenols present in tomatoes, and their peels, have been reported to possess anti-allergic properties [10]. However, it remains unclear whether tomatoes can prevent or exacerbate allergic diseases by regulating the number and/or function of eosinophils.

Thus, we examined the relationship between dietary tomato consumption and blood eosinophil count, as well as the change in this association based on baseline characteristics in middle-aged and older Japanese individuals.

## 2. Materials and Methods

### 2.1. Study Design and Participants

The data used in this cross-sectional study was sourced from the Shika study, a longitudinal observational community-based study that enrolls Shika residents and has been extensively described elsewhere [11,12,13,14,15,16]. In brief, this project linked to medical examination was conducted from 2013 in the town of Shika, located in the rural sector of Ishikawa Prefecture, Japan. The primary objective was to monitor the health conditions of the local resident aged 40 and above, and explore preventive measures for lifestyle-related diseases. The data included self-administered questionnaire and health examination data collected between 2013 and 2019 from the Shika study.

Figure 1 illustrates the study enrollment procedure. Among the 1335 participants aged ≥ 40 years who underwent a medical examination during the study period, ten were excluded for missing data on blood eosinophils and anthropometry. Furthermore, 38 participants who were receiving treatment for underlying diseases that could potentially impact the number of blood eosinophil counts were excluded. After excluding 262 participants who lacked data on their daily energy and tomato intake and 12 who reported a daily energy intake of <600 kcal/day or >4000 kcal/day, 1013 participants were included in the final analysis. The Shika study was approved by the Medical Ethics Committee at Kanazawa University (approval number: 1491). Written informed consent was obtained from all the participants.

### 2.2. Dietary Assessment of Tomato and Its Nutritional Components

Using the Japanese version of the food frequency questionnaire, a brief-type self-administered dietary history questionnaire (BDHQ), a standardized methodology was implemented to estimate nutritional intake from the data collected [17]. The validity of the BDHQ has been confirmed by other studies and is considered to have a satisfactory ranking ability for many nutrients among the Japanese population [18,19]. In this study, daily caloric and tomato intake and intake of the main vitamins in tomatoes (retinol equivalent, β-carotene equivalent, vitamin C, and folate) were estimated from the BDHQ results.

### 2.3. Measurement of Blood Eosinophil Counts

Peripheral blood eosinophil counts were measured by SRL, Inc. (Tokyo, Japan), a commercial clinical laboratory. Measurements were performed using an automated hematology analyzer, XN-9100+SP50 (Sysmex Corporation, Kobe, Japan), which allows simultaneous determination of total white blood cell counts and differential leukocyte subsets, including eosinophils.

### 2.4. Other Variables

Blood pressure (BP) was measured in the seated position with the individual at rest. Hypertension was defined as a BP ≥ 140/90 mmHg or the use of antihypertensive agents. BMI was calculated as weight (kg) divided by the square of height (m). Diabetes was defined as the use of antidiabetic medication, fasting plasma glucose ≥ 126 mg/dL, or HbA_1c_ ≥ 6.5%, the latter two of which are the cutoff values used by the Japan Diabetes Society to diagnose diabetes mellitus [20]. Dyslipidemia was defined based on the Japan Atherosclerosis Society guidelines as levels of triglycerides ≥ 150 mg/dL, HDL cholesterol < 40 mg/dL, and LDL cholesterol ≥ 140 mg/dL, or the use of lipid-lowering agents [21]. A urine dipstick test was performed using spot urine specimens, and proteinuria was classified as negative, 1+, 2+, 3+, and 4+, with a 1+ reading corresponding to a proteinuria of 30 mg/dL [22]. To evaluate kidney function in this Japanese population, eGFR was calculated as follows: eGFR (mL/min/1.73 m^2^) = 194 × serum creatinine^−1.094^ × age^−0.287^ (if female, × 0.739) [23]. CKD was defined as eGFR < 60 mL/min/1.73 m^2^ and/or proteinuria ≥ 1+ (corresponding to ≥300 mg/g) in accordance with the KDIGO 2012 Clinical Practice Guideline for the Evaluation and Management of Chronic Kidney Disease [24]. The enzymatic method was employed to measure serum creatinine levels. Asthma was defined as having been diagnosed by a doctor, while CVD was defined as a history of myocardial infarction, angina pectoris, stroke, or peripheral artery disease. These conditions were identified through self-reported questionnaires. Other variables, such as age, sex, smoking status, frequency of exercise, and alcohol consumption, were assessed using self-administered questionnaires. Smoking status was classified as either current or non-current (nonsmoker or past smoker), and drinking habits were defined as consuming more than one glass of Japanese sake (22 g ethanol) per day at least three times per week [15,25]. The exercise frequency was estimated by asking participants whether they had exercised for more than 30 min at least twice a week during the previous year or had performed tasks such as walking, cleaning, and carrying baggage for more than 1 h per day [15,25]. Participants with affirmative responses to either of these questions were considered to have completed an adequate amount of physical activity in accordance with the World Health Organization guidelines on physical activity [26].

### 2.5. Assessment of Genetic Variants

The genomic DNA was extracted using the services of either SRL, Inc. (Tokyo, Japan) or the QIAamp DNA Blood Maxi Kit (Qiagen, Hilden, Germany). Genome-wide genotyping was performed on 1325 blood samples using the Japonica Array v2 [27] (Toshiba Co. Ltd., Tokyo, Japan). Details of quality control procedures based on sex identity between karyotype and questionnaire, SNP call rates, individual call rates, Hardy–Weinberg equilibrium tests, inbreeding coefficients, cryptic relatedness, and population structure, and genotype imputation using the Beagle software version 4.1 [28,29] and the 1000 Genomes Project Phase 3 as a reference panel [30] were extracted in accordance with previously described procedures [31].

### 2.6. Calculation of Polygenic Risk Scores

To calculate polygenic risk scores (PRS) for asthma with LDPred [32], we downloaded the summary statistics of approximately 8.7 million SNPs that were examined in the genome-wide association study (GWAS) for asthma [33] from the Japanese encyclopedia of genetic associations by Riken (JENGER) database (http://jenger.riken.jp, accessed on 6 May 2024). The coordination step in the LDPred analysis resulted in the retention of 5,074,252 SNPs, which were subsequently used in the PRS calculation. Under the default settings of LDPred, we used the following values as the densities of SNPs associated with asthma: 1, 0.3, 0.1, 0.03, 0.01, 0.003, and 0.001. Based on the R^2^ values between the phenotypes for asthma (affected or unaffected) observed in 672 participants and their PRS estimated for each, we chose = 0.3 (R^2^ = 0.0121) as the most likely density of SNPs associated with asthma among the values tested in this study. The study sample-specific PRS was defined as the sum of the individual risk variant alleles, each multiplied by its corresponding weight based on the GWAS for asthma. An elevated PRS is indicative of an increased genetic susceptibility to asthma.

### 2.7. Statistical Analysis

The descriptive characteristics at baseline were compared in relation to the presence or absence of elevated blood eosinophil counts. Similar to a previous study [34], eosinophil counts were non-normally distributed with a right-skewed shape (Figure 2), and an elevated blood eosinophil count was defined as a value that exceeded the ≥75th percentile (≥204/μL). Student’s *t*-test was employed to compare continuous variables, while the chi-squared test was employed to compare categorical variables. Spearman’s rank correlation between dietary tomato intake and serum nonspecific IgE levels was examined.

The association between daily tomato consumption and elevated eosinophil counts was explored using a multivariable logistic regression analysis. In the analysis, each model was adjusted for potential confounders based on previous studies [34]. Model 1 was adjusted for age, sex, and daily energy intake, while model 2 was adjusted for BMI, drinking habits, current smoking status, and asthma, in addition to the variables considered in model 1. Furthermore, the PRS was incorporated as a covariate into model 2 for developing an exploratory model (model 3).

We performed the subgroup analysis to evaluate the interaction between dietary tomato intake and individual-level variables including age (<65 and ≥65 years), sex (male or female), BMI (BMI < 25 and ≥ 25 kg/m^2^), asthma, hypertension, dyslipidemia, diabetes, CKD, and CVD history. This allowed us to identify individuals for whom preventive measures are likely to be particularly effective.

All statistical analyses were conducted using SPSS software version 28 (IBM Corp., Tokyo, Japan). A two-tailed *p* < 0.05 was considered statistically significant.

## 3. Results

### 3.1. Participant Characteristics According to Blood Eosinophil Counts

Overall, 252 participants exhibited elevated blood eosinophil counts. Table 1 summarizes the participant characteristics and daily tomato intake according to blood eosinophil counts. Among the 1013 participants, the mean age was 62.5 (standard deviation, 11.2) years old, with 46.8% (*n* = 474) being male. Individuals in the high eosinophil group had a lower age than those in the normal eosinophil group. The elevated eosinophil group exhibited a higher BMI, prevalence of men, asthma, drinking habits, and number of current smokers compared to the normal eosinophil group. In addition to the proportion of individuals with asthma, the asthma-related PRS was higher in the elevated eosinophil group than in the normal eosinophil group. Although there was no difference in energy consumption, individuals with elevated eosinophil levels demonstrated a reduced daily intake of tomatoes compared to those without elevated eosinophil counts. There was no difference in comorbidity frequency among the groups.

### 3.2. Association Between Dietary Tomato Intake and Blood Eosinophil Count

The association between tomato consumption and eosinophil count is demonstrated in Table 2. After adjusting for confounding factors, an association was established between a low intake of tomatoes and an increased blood eosinophil count. Comparable results were obtained when asthma-related PRS was incorporated as a covariate in the exploratory model (Appendix A). Regarding the relationship with IgE, no correlation was observed with daily tomato intake (Appendix A).

### 3.3. Association Between Intake of Major Vitamins Contained in Tomatoes and Blood Eosinophil Count

Among the vitamins and minerals reported to be associated with blood eosinophil counts [35], the key ones contained in tomatoes are vitamin A, vitamin C, and folic acid [8,9]. After accounting for all potential confounders, no association between vitamin consumption and blood eosinophil count was identified when each vitamin was employed as an explanatory variable in place of tomato (Appendix A).

### 3.4. Subgroup Analysis by Participants’ Characteristics

Table 3 presents the subgroup analysis results based on the baseline status of participants, including age, sex, BMI, and comorbidities such as asthma. A significant relationship was observed between the presence or absence of CKD and dietary tomato intake (*p* for interaction = 0.019). However, there was no association between tomato consumption and an elevated eosinophil count in participants with CKD. The other subgroups did not exhibit any significant interaction.

## 4. Discussion

This study examined the relationship between the dietary intake of tomatoes and blood eosinophil count in middle-aged and older Japanese individuals. We discovered that low tomato intake was associated with high eosinophil counts, independent of other significant confounders. Furthermore, the subgroup analysis revealed that the association between dietary tomato intake and blood eosinophil counts was not significantly modified by the characteristics of the study participants.

The distribution of blood eosinophil counts was right-skewed in this study, and the 50th and 75th percentile values of the counts were 120 and 204 cells/µL, respectively. These findings were almost consistent with those observed in a previous population-based cohort study: the median (interquartile range) in the Copenhagen general population study was 170 (110–250) cells/µL [36], and the geometric mean and 75th percentile values in Austria were 128 and 210 cells/µL, respectively [34]. Furthermore, similar to a previous study [34], the variables male sex, asthma, overweight/obesity, and smoking status were associated with increased blood eosinophil counts. Regarding smoking status, the prevalence of smokers was double in subjects with high eosinophils compared to subjects with normal eosinophil count (31.3 vs. 15.4%). Smoking is a well-known risk factor for chronic obstructive lung diseases, including asthma and smoking is known to be associated with higher eosinophil count [37] and total IgE [38]. These factors might be linked to increased eosinophil levels that are prevalent in both Europeans and Asians. The results also indicate that, among these factors, management of overweight/obesity [39] and smoking cessation [40] are the typically essential lifestyle interventions beneficial for the prevention and management of allergic diseases involving eosinophils.

Despite adjusting for potential confounders, the present study demonstrated that a low dietary tomato intake was associated with an increased blood eosinophil count. In addition to vitamins and minerals, tomatoes contain functional components, including polyphenols and carotenoids such as lycopene, which are believed to exert antioxidant, anti-inflammatory, and anti-cancer effects [8,9]. Among these, flavonoids, a class of polyphenols contained in tomatoes, are known to have anti-allergic effects [6,10]. In particular, naringenin chalcone, which was isolated from tomato skin extract, has been reported to exert its anti-allergic effect by inhibiting histamine release from mast cells [41]. In a randomized clinical trial, it has been reported that oral administration of 360 mg/day of tomato extract enhanced nasal symptoms and QOL in patients with allergic rhinitis [42]. Combined with the present study findings, one potential mechanism for the anti-allergic properties of tomato may include the ability of its components to regulate pathologic conditions that involve eosinophils. Additional studies are required to elucidate these mechanisms in the future.

With the exception of CKD, the background characteristics of the study participants did not influence the relationship between dietary tomato intake and blood eosinophil count in the present study. This suggests that tomato intake can reduce blood eosinophil counts in middle-aged and older adults with diverse characteristics. Along with its health benefits, several adverse effects of tomato and tomato-based products have been reported, including gastroesophageal reflux disease, allergies, and kidney stones [8]. The differing relationship between tomato intake and blood eosinophil counts in individuals with and without CKD may be attributed to the following: In addition to the trend of avoiding consumption of raw tomatoes to prevent hyperkalemia in CKD patients, it is possible that cooking (or processing, when processed tomato-based products are consumed) removes the active ingredients that affect blood eosinophil levels along with potassium. As previously recommended [43,44,45], it may be essential to examine the health benefits and risks of potassium-rich foods, including tomatoes, in the dietary management of CKD patients. Furthermore, recent advances in understanding eosinophil biology have revealed that besides their involvement in the pathogenesis of allergic diseases and fibrosis, eosinophils play crucial roles in tissue homeostasis and repair [3,46]. Future studies on the potential of tomato consumption to prevent eosinophil-related diseases may require additional safety evaluations based on individual characteristics.

This study has several limitations. First, this was a cross-sectional study; therefore, a causal association could not be established between the intake of tomatoes and blood eosinophil counts. Therefore, it remains unclear whether dietary intake of tomatoes can prevent the elevation of blood eosinophil counts in the general population, including younger individuals. Regarding the study design, there is a further limitation that make the relationship between tomato consumption, eosinophilia and related T2 diseases insufficiently proven. Second, the BDHQ was used to collect data on dietary tomato intake, which may have been influenced by recall bias. Concomitantly, the forms of tomato intake (whether raw or processed) were unknown, despite the fact that different tomato varieties and their derived products contain a wide range of nutrient types and concentrations [8]. Finally, in addition to the fact that parasitic infections in the study participants were not excluded, which are a well-known cause of an increase in blood eosinophil counts, selection bias may have occurred because the Shika study participants were volunteers and thus more health-conscious than the general population.

## 5. Conclusions

In conclusion, this study demonstrated that low tomato intake was associated with elevated blood eosinophil counts in middle-aged and older persons and that the relationship was not influenced by the characteristics of the study participants. Additional basic and longitudinal studies are warranted to confirm this relationship and to determine whether specific dietary interventions with tomato can regulate blood eosinophil counts as biomarkers for diagnosis and evaluating treatment response of eosinophil-related disorders and type 2 inflammatory diseases.

## Figures and Tables

**Figure 1 nutrients-17-03467-f001:**
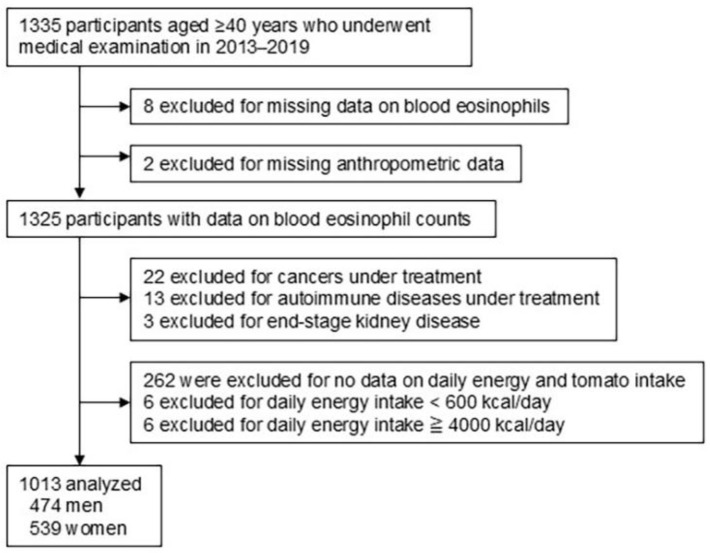
Flow diagram illustrating the enrollment procedure of the study participants.

**Figure 2 nutrients-17-03467-f002:**
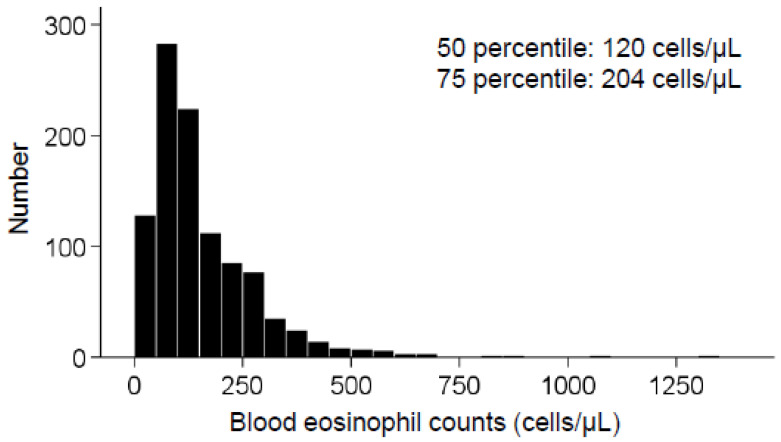
Distribution of blood eosinophil counts in the study cohort.

**Table 1 nutrients-17-03467-t001:** Participant demographics and corresponding blood eosinophil levels.

	Total (*n* = 1013)		Normal (*n* = 761)		High (*n* = 252)		*p*-Value
	Mean/*n*	SD/%	Mean/*n*	SD/%	Mean/*n*	SD/%	
Men, *n*	474	46.8	312	41.0	162	64.3	**<0.001**
Age, years	62.5	11.2	63.0	11.1	60.81	11.4	**0.007**
Asthma, *n*	36	3.6	20	2.6	16	6.3	**0.006**
Hypertension, *n*	582	57.5	431	56.6	151	59.9	0.361
Dyslipidemia, *n*	358	35.3	262	34.4	96	38.1	0.291
Diabetes, *n*	137	13.5	94	12.4	43	17.1	0.058
CVD, *n*	60	5.9	47	6.2	13	5.2	0.553
CKD, *n*	211	20.8	156	20.5	55	21.8	0.653
Current drinking, *n*	485	47.9	344	45.2	141	56.0	**0.003**
Current smoking, *n*	196	19.3	117	15.4	79	31.3	**<0.001**
Physical activities, *n*	572	56.5	434	57.0	138	54.8	0.529
BMI, kg/m^2^	23.4	3.2	23.2	3.1	23.9	3.5	**0.006**
White blood cell count, /μL	5804	1659	5501	1519	6721	1726	**<0.001**
Eosinophil count, /μL	155.0	126.7	98.3	48.3	326.3	135.9	**<0.001**
Eosinophil fraction, %	2.68	2.05	1.87	0.98	5.13	2.48	**<0.001**
IgE, IU/mL	219.1	573.3	169.5	319.8	362.7	980.0	**0.005**
PRS_asthma	0.267	0.090	0.261	0.089	0.283	0.092	**0.019**
Energy intake, kcal/day	1846.0	592.3	1828.8	592.5	1898.0	589.9	0.108
Tomato intake, g/day	18.4	25.3	19.8	27.0	14.1	18.8	**<0.001**

Data are expressed as *n*, % or mean, standard deviation. CVD, cardiovascular disease; CKD, chronic kidney disease; BMI, body mass index; IgE, immunoglobulin E; PRS, polygenic risk score. The *p*-values that are < 0.05 are highlighted in bold.

**Table 2 nutrients-17-03467-t002:** Multivariable logistic regression analysis results for elevated blood eosinophil counts.

	Model 1			Model 2		
	OR	95% CI	*p*-Value	OR	95% CI	*p*-Value
Men, vs. women	2.511	1.826–3.455	**<0.001**	2.100	1.449–3.044	**<0.001**
Age, per +1 year	0.983	0.970–0.996	**0.011**	0.986	0.973–1.000	0.051
Energy intake, per +1 kcal/d	1.000	1.000–1.000	0.815	1.000	1.000–1.000	0.716
Tomato intake, per +10 g/d	0.904	0.842–0.970	**0.006**	0.895	0.834–0.961	**0.004**
BMI, per +1 kg/m^2^				1.051	1.003–1.100	**0.037**
Drinking habit				0.930	0.663–1.303	0.672
Current smoking				1.818	1.260–2.622	**0.001**
Asthma				3.238	1.590–6.595	**0.001**

Forced entry binary multivariable logistic regression models were implemented (*n* = 1013). OR, odds ratio; CI, confidence interval; BMI, body mass index. The *p*-values that are <0.05 are highlighted in bold.

**Table 3 nutrients-17-03467-t003:** Subgroup analysis of the association between tomato consumption and increased eosinophil count.

	OR	95% CI		*p*-Value	*p*-Value for Interaction
Sex					0.915
Men	0.904	0.825	1.000	**0.042**	
Women	0.877	0.776	0.990	**0.034**	
Age					0.995
<65 y	0.877	0.784	0.970	**0.013**	
≥65 y	0.923	0.834	1.030	0.148	
Asthma					0.571
No	0.886	0.825	0.961	**0.004**	
Yes	0.942	0.722	1.219	0.650	
BMI					0.641
<25 kg/m^2^	0.886	0.809	0.970	**0.011**	
≥25 kg/m^2^	0.914	0.792	1.051	0.205	
Hypertension					0.662
No	0.851	0.745	0.970	**0.016**	
Yes	0.923	0.842	1.020	0.111	
Dyslipidemia					0.897
No	0.904	0.825	1.000	**0.045**	
Yes	0.868	0.768	0.980	**0.025**	
Diabetes					0.940
No	0.904	0.834	0.990	**0.022**	
Yes	0.801	0.638	0.990	**0.045**	
CKD					**0.019**
No	0.851	0.768	0.932	**<0.001**	
Yes	1.083	0.942	1.231	0.270	
CVD					0.672
No	0.886	0.817	0.961	**0.004**	
Yes	0.932	0.708	1.219	0.595	

Covariates: age, sex, daily energy intake, BMI, drinking habits, current smoking and asthma. OR, odds ratio; CI, confidence interval; BMI, body mass index; CKD, chronic kidney disease; CVD, cardiovascular disease. The *p*-values that are <0.05 are highlighted in bold.

## Data Availability

The data that support the present study’s findings are available from the corresponding author upon reasonable request, while they are not publicly available due to privacy and ethical policies.

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
