# Peer review of "Association Between Dietary Tomato Intake and Blood Eosinophil Count in Middle-Aged and Older Japanese Individuals: A Population-Based Cross-Sectional Study [Author-notes fn1-nutrients-17-03467]"

_nutrients, 2025, doi:10.3390/nu17213467_

Round 1

Reviewer 1 Report

Comments and Suggestions for Authors

With real interest, I read the manuscript entitled ”Association between dietary tomato intake and blood eosinophil count in middle-aged and older Japanese: a population-based cross-sectional study” (Manuscript ID: nutrients-3902607), written by Hara and colleagues. Overall, it is an elegantly written paper, based on a solid study. I have the following minor comments/suggestions:

1.        The whole genetic (polygenic score) appears in this work very late, and it is not even mentioned in the abstract or introduction, if I haven overlooked anything. How important is this genetic aspect for the story? Should it be more highlighted or rather not?

2.        It is not fully clear to me which parts of this research were performed exclusively for the purposes of the present paper, and which are reused after the past studies?

3.        Please, mention that in general the interaction between the environment and genetic background, often mediated by the epigenetic mechanisms, is involved in exerting the dietary effects on allergies development (PMID: 33668787).

4.        Graphical abstract would further increase already very height elegance of this paper.

Reviewer 2 Report

Comments and Suggestions for Authors

My comments:

1. The title states that the patients are middle-aged. However, data from patients aged 40 and over were analyzed, and no upper age limit was specified. Furthermore, data from patients aged 65 and over were also analyzed. The nature of this group was not specified. Why was this analyzed? If there is a reference group, why wasn't a group younger than 40 also identified?

2. The methodology for performing many different analyses was described, but the purpose of these analyses was not specified. What do they have to do with the topic of the manuscript? Why were blood pressure, renal function, lipid profile, etc. analyzed?

3. The method of eosinophil count analysis was not specified. What analytical system was used, or perhaps a manual technique? And who performed it?

4. How was tomato consumption analyzed? Where did the data come from?

5. Was only raw tomato consumption analyzed, or also processed tomato consumption? This should be separated due to the fact that tomato processing significantly affects the lycopene concentration in the product, which may be important in the context of these analyses.

6. The introduction does not explain why the authors chose eosinophils as an indicator of allergic inflammation. In my opinion, more than one inflammatory effector should be analyzed for the results of these analyses to be reliable.

7. The conclusions are very general and do not comprehensively reflect the nature of the allergic reaction. We do not know how tomatoes affect the various inflammatory pathways involved in the allergic process. Such fragmentary studies may lead to a distorted perception of immunological processes and, consequently, be harmful to the patient.

Reviewer 3 Report

Comments and Suggestions for Authors

Authors explored the association between dietary tomato intake and blood eosinophil count in Japanese individuals. At this aim they performed a population-based, cross-sectional study which included 1,013 participants aged≥40 years. The dietary intake of tomatoes was assessed using a validated,  self-administered diet history questionnaire. They found that an increase in tomato intake of 10 g  was inversely associated with an elevated blood eosinophil count (OR, 0.895; 95% CI, 0.834–0.961), defined as as a value that exceeded the≥75th percentile (≥204/μL). Authors concluded that their results may provide insight into the dietary management of eosinophil-related allergic and type 2  inflammatory diseases.

The following points need to be addressed in a revised version of the manuscript.

1-The prevalence of smokers is double in subjects with high eosinophils compared to subjects with normal eosinophil count (31.3 vs 15.4 %). Smoking is a well known risk factor for chronic obstructive lung diseases, including asthma and smoking is known to be associated with higher eosinophil count  (Benson VS, Hartl S, Barnes N, et al. Blood eosinophil counts in the general population and airways disease: a comprehensive review and meta-analysis. Eur Respir J 2022) and total IgE ( Relationships between total serum IgE, atopy, and smoking: A twenty-year follow-up analysis. Sherrill, Duane L. et al. Journal of Allergy and Clinical Immunology, 1994)

2-It is not clear why Authors included only people > 40 years, when it is known that allergic diseases are particularly prevalent in younger population.

3-Authors considered only asthma as T2 inflammatory diseases, ignoring other more frequent allergic diseases such as alletgic rhinitis. Moreover, asthma may be sustained by non T2 mechanisms.

Round 2

Reviewer 2 Report

Comments and Suggestions for Authors

I thank the authors for their clarifications and improvements to the manuscript. There are no new comments.

Reviewer 3 Report

Comments and Suggestions for Authors

The authors have modified the manuscript taking into account the criticisms and suggestions, but unfortunately important limitations of the study remain, limitations that make the relationship between tomato consumption, eosinophilia and related T2 diseases insufficiently proven.
